# DISTRIBUTIONAL MONTE–CARLO TREE SEARCH WITH THOMPSON SAMPLING IN STOCHASTIC ENVIRONMENTS

## ABSTRACT

Monte–Carlo Tree Search (MCTS) excels in deterministic settings, but can struggle in highly stochastic domains where transition randomness and partial observability induce underexploration and miscalibrated value estimates. We integrate *distributional* Reinforcement Learning (RL) with Thompson Sampling (TS) plus an optimistic exploration bonus, yielding two *distributional MCTS* algorithms: CATSO (Categorical TS with Optimism) and PATSO (Particle TS with Optimism). Each Q–node maintains a return *distribution*—via fixed atoms (CATSO) or dynamically capped particles with *merge-on-insert* (PATSO). Actions are chosen by a Thompson draw *plus* a polynomial optimism term. We prove both algorithms attain non–asymptotic, problem–dependent *simple regret* $O(n^{-1/2})$ at the root. We further connect the resulting tree policies to *Wasserstein* Distributionally–Robust MDPs (WDRMDPs), giving a robustness interpretation and a sample–complexity bound. Experiments on synthetic stochastic trees and 12 Atari games indicate improved robustness under uncertainty relative to non-distributional baselines. We provide comprehensive ablations isolating the contributions of distributional Q-nodes, Thompson sampling, optimism, and backup rules, alongside hyperparameter sensitivity analysis and runtime/memory measurements.

## 1 INTRODUCTION

Online planning in Markov Decision Processes (MDPs) involves making real-time decisions balancing exploration and exploitation under uncertainty. Monte–Carlo Tree Search (MCTS) is a powerful framework that has achieved remarkable success in games (Silver et al., 2016; 2017; Schrittwieser et al., 2020), robotics, and control tasks. However, classical MCTS variants—which typically back up *scalar* mean values at Q–nodes and use UCB–style bonuses (e.g., UCT (Kocsis et al., 2006))—can struggle in highly stochastic environments where transition randomness leads to biased value estimates, underexploration, and premature pruning of promising actions.

### 1.1 OUR APPROACH AND CONTRIBUTIONS

We endow each Q–node with a *distribution* over returns and select actions using *Thompson Sampling* (to exploit information about the full shape of return distributions) *plus* an *optimistic* count–based bonus (to guard against posterior miscalibration). We instantiate this design with two parameterizations: CATSO, which uses categorical atoms with a Dirichlet posterior, and PATSO, which uses an online particle approximation with a cap-and-merge memory scheme.

**Core contributions.** Our main contributions are: (1) Distributional Q–nodes + TS + optimism for MCTS: we maintain *full return distributions* at Q–nodes and select actions via Thompson sampling plus a polynomial optimism bonus (Sec. 3). (2) Bandit-to-MCTS analysis for distributional TS: we model each node as a non-stationary bandit, prove $O(n^{-1/2})$ simple regret for CATSO/PATSO, and lift this to the full tree (Secs. 4.1–4.2). (3) Memory-capped PATSO with guarantees: a merge-on-insert scheme caps particles at $K$ and adds only $O(\text{range}/(K(1-\gamma)^2))$ simple regret (Sec. 4.4). (4) WDRMDP connection: Q-node distributions induce Wasserstein balls around empirical return laws; the resulting policy is near-optimal for a WDRMDP with a sample-complexity bound (Sec. 4.3). (5) Empirical validation and ablations: on synthetic trees and 12 Atari games we isolate the roles of

distributional Q-nodes, TS vs. UCB, backup rules, stochasticity, and hyperparameters, and report runtime/memory (Sec. 5).

**What is new.** While power-mean backups, polynomial optimism, and Thompson sampling each appear in prior work, to our knowledge we are the first to unify (i) distributional Q-nodes (categorical or particle), (ii) Thompson sampling at Q-nodes, and (iii) polynomial optimism within a single MCTS framework, providing both non-asymptotic guarantees and a novel WDRMDP interpretation

**Why V–nodes remain scalar.** Propagating full distributions at V–nodes would require repeated cross-child convolutions, inflating compute/memory exponentially with branching factor and depth, and complicating both implementation and analysis. We therefore keep Q–nodes distributional (where action decisions are made) and use a *power–mean* backup over child *means* at V–nodes to interpolate between averaging and max (Dam et al., 2019; 2024). Section 3.3 clarifies this design choice, and our ablations (Sec. 5.3) isolate the role of the backup rule.

## 2 SETTING AND BACKGROUND

We consider an infinite-horizon discounted MDP $\mathcal{M} = \langle \mathcal{S}, \mathcal{A}, \mathcal{R}, \mathcal{P}, \gamma \rangle$ with state space $\mathcal{S}$, action set $\mathcal{A}$, transition distribution $\mathcal{P}(\cdot|s,a)$, discount factor $\gamma \in (0,1]$, and bounded rewards $\mathcal{R}(s,a,s') \in [0, R_{\max}]$. The optimal action-value function $Q^\star$ satisfies the Bellman equation:

$$Q^\star(s,a) = \mathbb{E}_{s' \sim \mathcal{P}(\cdot|s,a)}\big[\mathcal{R}(s,a,s') + \gamma \max_{a'} Q^\star(s',a')\big].$$

**MCTS planning framework.** MCTS plans from initial state $s_0$ by repeatedly simulating trajectories of depth $H$ (for analysis) under a fixed playout policy $\pi_0$ with value function $V_0$. This produces estimates $\widehat{Q}_n(s_0,a)$ of root action-values. Let $V^\star(s_0) = \max_a Q^\star(s_0,a)$ and $\widehat{V}_n(s_0) = \max_a \widehat{Q}_n(s_0,a)$. The *simple regret* at time $n$ is:

$$R(s_0, n) = V^\star(s_0) - \widehat{V}_n(s_0).$$

For node $s_h$ at depth $h \leq H-1$ from root $s_0$, we define estimated value functions inductively: $\widehat{V}(s_H) = V_0(s_H)$ at depth $H$, and for $h \leq H-1$:

$$\widetilde{Q}(s_h, a) = r(s_h, a) + \gamma \sum_{s_{h+1} \in \mathcal{S}} \mathcal{P}(s_{h+1}|s_h, a)\widetilde{V}(s_{h+1}),$$

$$\widetilde{V}(s_h) = \max_a \widetilde{Q}(s_h, a),$$

where $r(s_h, a) = \mathbb{E}[\mathcal{R}(s_h, a, s')]$. This yields the approximation bound:

$$|Q^\star(s_0, a) - \widetilde{Q}(s_0, a)| \leq \gamma^H \|V^\star - V_0\|_\infty.$$

MCTS minimizes simple regret by accurately estimating $\widetilde{Q}(s_0, a)$ and $\widetilde{V}(s_0)$ to approximate $V^\star(s_0)$, $Q^\star(s_0, a)$, and identify the optimal action $a_\star = \arg\max_a Q^\star(s_0, a)$.

**Distributional view.** Formally, we view each Q-value as a distribution over possible returns. Letting $\mathcal{X}(s,a)$ denote the immediate reward distribution:

$$\mathcal{Q}(s,a) \overset{D}{=} \mathcal{X}(s,a) + \gamma\,\mathcal{V}(s'),$$

where $s' \sim \mathcal{P}(\cdot|s,a)$ and $\mathcal{V}(s')$ merges the Q distributions under the tree policy. We track a distribution $\mathcal{Q}(s,a)$ at each Q-node rather than a scalar mean.

## 3 DISTRIBUTIONAL THOMPSON SAMPLING WITH OPTIMISM IN TREE SEARCH

A pure Thompson Sampling (TS) approach can *under*estimate the value of less-visited actions, leading to insufficient exploration in stochastic environments. Several works have proposed *combining*

TS with an explicit optimism bonus or confidence bound: *Optimistic Thompson Sampling* (May et al., 2012) adds a UCB-style term to the TS-sampled value in multi-armed bandits; *Bayes-UCB* (Kaufmann et al., 2012) selects actions by the upper quantile of a Bayesian posterior; *PSRL + Optimism* (Osband et al., 2013; 2018) incorporates additional confidence bonuses into posterior sampling. In MCTS, Bai et al. (2013; 2014) investigated Bayesian model adaptation with variance-based exploration. These lines of work share a principle: *augmenting Thompson-drawn estimates with a boost for rarely visited actions* enhances exploration if the posterior is temporarily miscalibrated.

Building on these ideas, we propose two *distributional* MCTS algorithms combining TS and an optimism bonus:

- CATSO (Categorical Thompson Sampling with Optimistic Bonus)
- PATSO (Particle Thompson Sampling with Optimistic Bonus)

Both maintain a *distribution* over Q-values at each node, use Thompson sampling to select actions, and then *add* a polynomial exploration bonus. This synergy captures uncertainty in return estimates while preventing underexploration.

### 3.1 CATSO: CATEGORICAL TS + OPTIMISM

Each Q–edge $(s, a)$ tracks $N$ uniformly spaced atoms $z_i$ on a dynamic interval $[Q_{\min}, Q_{\max}]$, together with Dirichlet counts $\alpha^i$ whose normalized version defines the categorical probabilities.

Initially, $Q_{\min} = 0$ and $Q_{\max} = 0.001$. When a new return $\overline{Q}_t(s, a) = r_t + \gamma \widehat{V}(s')$ arrives:

Each Q-edge $(s, a)$ maintains **atom locations** $\{z_i(s, a)\}_{i=0}^{N-1}$ (uniform in $[Q_{\min}, Q_{\max}]$), **categorical probabilities** $\{p_i(s, a)\}_{i=0}^{N-1}$ over these atoms, and a **Dirichlet prior** $\mathrm{Dir}(\alpha^0(s, a), \ldots, \alpha^{N-1}(s, a))$ for Thompson sampling.

### 3.2 PATSO: PARTICLE TS + OPTIMISM (CAPPED, DEFAULT)

Each Q-edge $(s, a)$ maintains a sorted list of distinct returns $\mathcal{S}(s, a)$ (particles) with weights $\alpha(s, a)$; on a new sample $\overline{Q}_t$ we either increment the weight of an existing particle (if $\overline{Q}_t \in \mathcal{S}(s, a)$) or insert a new particle with weight 1.

To bound memory, we cap the particle set at size $K$ via *merge-on-insert*: if the set is full, merge the closest neighboring particles (preserving the first moment) before inserting the new particle.

**Action selection.** At node $s_h$, for each action $a$ we draw $L(s_h, a) \sim \mathrm{Dir}(\alpha(s_h, a))$ and compute a Thompson value $\phi(s_h, a) = \sum_{z \in \mathcal{S}(s_h, a)} z \, L_z(s_h, a)$. We then add the optimism bonus $B(n, s_h, a) = C \, T_{s_h}(n)^{1/4} / T_{s_h, a}(n)^{1/2}$ to obtain $\phi'(s_h, a) = \phi(s_h, a) + B(n, s_h, a)$, and select the action $a^\star = \arg\max_a \phi'(s_h, a)$.

**Remark 1** (CATSO vs. PATSO). *CATSO discretizes Q-values into $N$ fixed atoms, potentially introducing approximation error if $N$ is small. PATSO grows its particle set automatically, yielding a finer representation. However, carefully choosing $N$ in CATSO can be more memory-efficient, whereas PATSO's flexibility requires the cap-and-merge mechanism. Both share similar theoretical guarantees.*

### 3.3 VALUE BACKUP AT V–NODES

Although Q-nodes track full distributions, V-nodes store only a *power-mean backup* of child Q-means:

$$\widehat{V}(s) = \left( \sum_a \frac{T_{s,a}(n)}{T_s(n)} \, [\widehat{Q}(s, a)]^p \right)^{1/p}, \quad p \geq 1.$$

This interpolates between plain averaging ($p = 1$) and a maximum operator ($p \to \infty$) (Dam et al., 2019; 2024). In CATSO, $\widehat{Q}(s, a)$ is the expected value of the categorical distribution; in PATSO, it is the weighted average of all particles.

**Design rationale.** Extending full distributions to V-nodes would require repeated convolutions across all child distributions, making both analysis and computation significantly more complex (potentially exponential in branching factor and depth). By keeping distributions only at Q-nodes (where decisions are made) and using the power-mean backup at V-nodes, we retain the core novelty—distributional decision-making at Q-nodes—while keeping the framework tractable. Our ablations (Sec. 5.3) isolate the effect of different backup rules.

## 4 THEORETICAL ANALYSIS

Our goal is to prove that both `CATSO` and `PATSO` achieve simple regret rate $O(n^{-1/2})$ under suitable assumptions. We begin by modeling each MCTS node as a *non-stationary multi-armed bandit*, establish convergence results in this setting, then extend them to the full tree via dynamic programming.

### 4.1 NON-STATIONARY BANDIT SETTING

Consider a bandit with $K$ arms. Each arm $a$ has a possibly non-stationary reward process $\{R_{a,t}\}$ taking values in $[0, R_{\max}]$. Let $\widehat{\mu}_{a,n}$ denote the empirical mean reward of arm $a$ after $n$ pulls. We assume the reward process stabilizes: $\mu_a = \lim_{n\to\infty} \mathbb{E}[\widehat{\mu}_{a,n}]$.

We define a *power mean* over the empirical means: $\widehat{\mu}_n(p) = \left( \sum_{a=1}^{K} \frac{T_a(n)}{n} \left[ \widehat{\mu}_{a,T_a(n)} \right]^p \right)^{1/p}$, where $T_a(n)$ counts how many times arm $a$ has been chosen in $n$ total rounds.

**Definition 1.** *A sequence of estimators* $(\widehat{V}_n)_{n\geq 1}$ *converges to $V$ with* concentration *if:*

- *(A) Concentration: For every $n \geq 1$ and $\varepsilon > 0$, there exists $c > 0$ such that*

$$\mathbb{P}(|\widehat{V}_n - V| > \varepsilon) \leq c \, \frac{1}{n\varepsilon^2}.$$

- *(B) Convergence:* $\lim_{n\to\infty} \mathbb{E}[\widehat{V}_n] = V$.

*We write* $\plim_{n\to\infty} \widehat{V}_n = V$ *when both hold.*

**Assumption 1** (Asymptotic stationarity with concentration). *For each arm $a \in [K]$, the empirical means $\widehat{\mu}_{a,n}$ satisfy* $\plim_{n\to\infty} \widehat{\mu}_{a,n} = \mu_a$.

**Justification in MCTS.** In our MCTS setting, each node's reward is a bounded finite-horizon return under a fixed playout policy. Non-stationarity arises from the tree expanding: early in the search, some rollouts see a truncated subtree; later, more siblings and deeper nodes are explored. Importantly, the expansion process stabilizes: for any fixed node, after sufficient depth and visits, the local subtree stops changing significantly, and subsequent returns are i.i.d. samples from a limiting distribution (the truncated value $\widetilde{Q}(s_h, a)$). Under bounded rewards and fixed playout, the induced reward process at each node is *asymptotically stationary*, satisfying Assumption 1. This is analogous to assumptions in Shah et al. (2022) for Fixed-Depth-MCTS.

**Theorem 1** (`CATSO` in Non-Stationary Bandits). *Let $(\widehat{\mu}_{a,n})_{n\geq 1}$ be bounded in $[0, R_{\max}]$ and satisfy Assumption 1, with $\mu_\star = \max_a \mu_a$. Assume `CATSO` (the categorical TS approach) initially samples each arm once, then follows the exploration strategy in Sec. 3.1. For any $p \geq 1$, the power-mean estimator $\widehat{\mu}_n(p)$ converges to $\mu_\star$ in the sense of Definition 1:*

$$\plim_{n\to\infty} \widehat{\mu}_n(p) = \mu_\star.$$

**Theorem 2** (`PATSO` in Non-Stationary Bandits). *Under the same assumptions as Theorem 1, if `PATSO` (the particle TS approach) is employed, the identical convergence result holds:*

$$\plim_{n\to\infty} \widehat{\mu}_n(p) = \mu_\star.$$

Proofs of both theorems are in Appendix A. The key insight is that the combination of Thompson sampling and the polynomial optimism bonus ensures sufficient exploration of all arms, while the distributional representation allows the algorithm to distinguish between actions with similar means but different tail behavior.

## 4.2 FROM BANDITS TO MCTS

Within MCTS, each internal node can be viewed as a non-stationary bandit: as the search tree grows, the empirical rewards from each node's children evolve. By applying Theorems 1 and 2 at every internal node, we show that each Q-node value concentrates around its truncated optimal value $\widetilde{Q}(s_h, a)$, and thus $\widetilde{V}(s_h)$ converges accordingly.

**Theorem 3** (Convergence of CATSO). *At the root node $s_0$, there exists a constant $C' > 0$ such that:*

$$\mathbb{E}[|\widehat{V}_n(s_0) - \widetilde{V}(s_0)|] \leq C' \, n^{-1/2}.$$

**Theorem 4** (Convergence of PATSO). *At the root node $s_0$, there exists a constant $C' > 0$ such that:*

$$\mathbb{E}[|\widehat{V}_n(s_0) - \widetilde{V}(s_0)|] \leq C' \, n^{-1/2}.$$

**Remark 2.** *For any power-mean exponent $p \geq 1$, both CATSO and PATSO inherit the same $O(n^{-1/2})$ rate seen in Stochastic-Power-UCT (Dam et al., 2024). This encompasses well-known backup rules like max (when $p \to \infty$) or mean (when $p = 1$). Under mild assumptions (bounded rewards, finite state space, fixed depth $H$), the constant $C'$ scales at most polynomially in $|\mathcal{S}|$, $|\mathcal{A}|$, and $H$ (see Appendix F for details).*

Thus, both distributional Thompson sampling approaches achieve a convergence rate matching the best known results for fixed-depth MCTS (Shah et al., 2022), bridging the analysis of non-stationary bandits to the multi-level structure of tree search.

## 4.3 CONNECTION TO WASSERSTEIN DISTRIBUTIONALLY ROBUST OPTIMIZATION

The distributional MCTS approach extends naturally to robust planning under uncertainty. We establish formal connections between our algorithms and Wasserstein Distributionally Robust Optimization (WDRO).

Let $\widehat{P}$ denote the empirical distribution of returns at a Q-edge (represented by categorical atoms in CATSO or particles in PATSO). Define the Wasserstein ball:

$$\mathcal{B}_\varepsilon(\widehat{P}) = \{P : W_p(P, \widehat{P}) \leq \varepsilon\},$$

where $W_p$ is the $p$-Wasserstein distance. We consider the Wasserstein Distributionally Robust MDP (WDRMDP):

$$\pi^\star \in \arg\max_\pi \min_{P \in \mathcal{B}_\varepsilon(\widehat{P})} \mathbb{E}_P[V^\pi].$$

This formulation seeks policies that are optimal under the worst-case perturbation of the empirical distribution within a Wasserstein ball. Our concentration results imply that the empirical distributions at Q-nodes concentrate around the true distributions in Wasserstein distance, and by the Lipschitz continuity of the Bellman operator with respect to $W_1$, the resulting tree policy is near-optimal for the WDRMDP.

**Theorem 5** (Robust Planning Guarantee). *Let $\mathcal{M}$ be an MDP with state space $\mathcal{S}$, action space $\mathcal{A}$, bounded rewards in $[0, R_{\max}]$, and discount factor $\gamma$. The number of samples required to learn an $(\varepsilon, \delta)$-robust policy using CATSO/PATSO combined with a concentration-based Wasserstein radius is:*

$$O\left(\left[\frac{R_{\max}^3}{\varepsilon(1-\gamma)^3} \log\left(\frac{H|\mathcal{S}|^2|\mathcal{A}|}{\delta}\right)\right]^{2H}\right).$$

This result shows that CATSO/PATSO do not merely estimate values—they implicitly construct robust policies. The Wasserstein ball radius shrinks at rate $O(n^{-1/2})$, ensuring that as simulations increase, the policy converges to optimality for both the nominal and perturbed MDPs.

**Remark 3** (Robust Planning via Distributional Estimation). *The connection to WDRO reveals that our distributional MCTS algorithms naturally provide robustness against model uncertainty. By maintaining and updating full distributions rather than point estimates, CATSO and PATSO implicitly construct uncertainty sets formalizable within the Wasserstein framework. The exponential dependence on horizon $H$ is typical for finite-horizon robust optimization problems, while the polynomial dependence on $|\mathcal{S}|$, $|\mathcal{A}|$, and $R_{\max}$ demonstrates computational tractability for moderate-sized problems.*

A detailed proof is provided in Appendix B. The intuition is that the empirical return distribution at each edge defines a Wasserstein ball around the true distribution, and optimizing with respect to these balls yields policies robust to model misspecification.

## 4.4 MEMORY AND COMPUTATION

**CATSO complexity.** Each $(s, a)$ edge stores $N$ atom locations and their Dirichlet parameters, requiring $O(N)$ memory per Q-edge. Selection involves sampling from a Dirichlet distribution and computing a dot product, costing $O(N)$ per decision. In the worst case, if one expands a full $|\mathcal{S}| \times |\mathcal{A}|$ frontier, memory scales as $O(N|\mathcal{S}||\mathcal{A}|)$. In practice, MCTS is budget-limited and biased toward promising branches, so the number of distinct Q-edges visited is $O(n)$ for $n$ rollouts, keeping memory linear in the explored tree size.

**PATSO memory management (cap with merge-on-insert).** Without intervention, `PATSO`'s particle set on a Q-edge may grow unboundedly. We cap it at $K$ particles via *merge-on-insert*:

1. Maintain a *sorted* list of pairs $\{(Q_i, \alpha_i)\}_{i=1}^M$ (value, weight) with $M \leq K$.
2. When a new sample $z$ arrives:
   (a) If $z$ matches an existing value (within numerical tolerance): increment its weight.
   (b) Else if $M < K$: insert $(z, 1)$ in sorted order.
   (c) Else ($M = K$; overflow): merge the closest neighboring pair (preserving first moment), then insert $(z, 1)$.

**Guarantees of the cap.** Let $\text{range}(s, a) = Q_{\max}(s, a) - Q_{\min}(s, a)$ be the local return range on a Q-edge. Nearest-neighbor merge preserves the *first moment* exactly and increases the $W_1$ distance to the uncapped empirical distribution by at most the merged gap times the merged mass. With a size cap $K$ and nearest-neighbor merges, the total discrepancy satisfies: $W_1(\widehat{P}, \widetilde{P}) = O\left(\frac{\text{range}(s,a)}{K}\right)$. Because the Bellman operator is $1/(1-\gamma)$-Lipschitz in $W_1$, the cap induces at the root an *additive* value error: $O\left(\frac{\text{range}}{K(1-\gamma)}\right)$ and thus in simple regret $O\left(\frac{\text{range}}{K(1-\gamma)^2}\right)$. Combining with the baseline $O(n^{-1/2})$ rate yields: simple regret $= O(n^{-1/2}) + O\left(\frac{\text{range}}{K(1-\gamma)^2}\right)$.

**Practical choice of $K$.** We set $K$ so that the additive term is below the statistical term at our rollout budgets. For example, choose $K \propto \sqrt{n}$ or pick the smallest $K$ with $\text{range}/(K(1-\gamma)^2) \ll n^{-1/2}$. In practice, moderate values like $K = 100$ or $K = 200$ suffice for typical MCTS budgets ($n = 1000$–$10000$ rollouts per decision).

**Complexity with the cap.**

| Operation | Cost |
|---|---|
| Per-$(s, a)$ memory | $2K$ floats $\Rightarrow O(K)$ |
| Per-selection compute | Dirichlet draw over $K$ bins + dot product $\Rightarrow O(K)$ |
| Insertion (incl. merge) | Binary search $O(\log K)$ + merge $\Rightarrow O(\log K)$ |

`CATSO` has fixed $O(N)$ per-edge memory and $O(N)$ selection compute. `PATSO` with capping has $O(K)$ per-edge memory, $O(K)$ selection compute, and $O(\log K)$ amortized insertion. The cap keeps memory bounded while preserving the $O(n^{-1/2})$ convergence rate up to a tunable additive term that vanishes as $K$ grows.

## 5 EXPERIMENTS

We compare `CATSO` and `PATSO` with several baselines: **UCT** (Kocsis et al., 2006), the classical MCTS baseline; **Fixed-Depth-MCTS** (Shah et al., 2022), a state-of-the-art method with non-asymptotic guarantees; **MENTS**, **RENTS**, and **TENTS** (Xiao et al., 2019; Dam et al., 2021), which use entropy-regularized tree search; **BTS** (Painter et al., 2024), a Bayesian tree-search method; **DNG** (Bai et al., 2013), which uses a Gaussian approximation to model return distributions; and

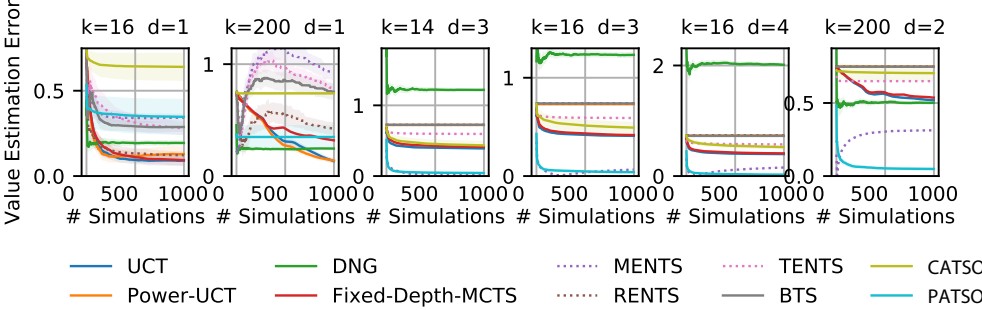

Figure 1: **Performance in the SyntheticTree environment.** We plot the absolute error of the estimated root value vs. the number of simulated trajectories $n$. Both CATSO and PATSO converge faster than non-distributional baselines, showcasing their robustness in stochastic transitions.

**Power-UCT** (Dam et al., 2024), which augments UCT with a power-mean backup similar to ours but without distributional Q-nodes.

We first evaluate on a *synthetic tree* environment, then on 12 Atari games. In all experiments, we set $\gamma = 1$ for synthetic and $\gamma = 0.99$ for Atari, using $N = 100$ atoms for CATSO and $K = 200$ particles (capped) for PATSO.

We include a range of baselines to clarify what each component of our approach contributes. UCT serves as the foundational MCTS method. Fixed-Depth-MCTS is our main theoretical reference point, since it attains the same $O(n^{-1/2})$ simple-regret rate that we target. MENTS, RENTS, and TENTS represent entropy-regularized tree search under stochasticity and illustrate the trade-off between optimizing a regularized objective and the true return. BTS and DNG are Bayesian or distributional MCTS variants that are closest in spirit to our work, allowing comparison against alternative distributional or posterior-sampling strategies. Finally, Power-UCT uses the same power-mean backup as our algorithms but combines it with a UCB-style exploration term instead of distributional Q-nodes, which lets us specifically isolate the impact of modeling full return distributions at Q-nodes.

## 5.1 SyntheticTree Environment

The SyntheticTree (Dam et al., 2021) has depth $d$, branching factor $k$, and stochastic edges. Each leaf node's reward is sampled from Gaussian($\mu$, $\sigma = 0.5$) with $\mu \in [0, 1]$. Transitions are stochastic: with 50% probability the agent moves to the intended child, otherwise transitions uniformly among other children. The goal is to find the leaf with highest mean reward. We test $(k, d)$ in $\{2, 4, 6, 8, 10, 12, 14, 16, 100, 200\} \times \{1, 2, 3, 4\}$, repeating each combination over 10 seeds.

Figure 1 shows that both CATSO and PATSO converge more rapidly than baselines in terms of root-value error. PATSO often outperforms CATSO because it does not rely on a fixed-atom approximation, but theoretical convergence rates are identical. The gap between our methods and baselines widens as stochasticity increases, consistent with our hypothesis that distributional representations better capture uncertainty in highly random environments.

## 5.2 Atari Experiments

We evaluate on 12 Atari games using a pretrained DQN network as a feature extractor for MCTS variants, following Xiao et al. (2019); Dam et al. (2021). We run 1000 simulations per decision step. Each method is evaluated over 10 episodes per game, using 3 different random seeds for a total of 30 evaluation episodes per game. Table 1 summarizes mean returns $\pm$ standard deviation.

**Performance analysis.** Our proposed algorithms demonstrate strong performance across the test suite. CATSO achieves the highest scores in 4 games and ties for best in 4 others; PATSO leads in 5 games and ties in 3 others. The performance gap is most significant in games requiring complex exploration—for instance, in Frostbite, PATSO outperforms MENTS by $17.9\times$ and TENTS by $7.4\times$.

Table 1: Mean returns $\pm$ standard deviation on 12 Atari games (higher is better). Best per game in **bold**. The final "Wins/Ties" row counts, for each method, the number of games where it achieves (i) the highest mean return (wins) or (ii) ties with another method for the highest mean return (ties).

| Game | `CATSO` | `PATSO` | MENTS | RENTS | TENTS | UCT |
|---|---|---|---|---|---|---|
| Alien | $1124 \pm 154$ | $\mathbf{1980 \pm 539}$ | $238 \pm 62$ | $326 \pm 116$ | $1260 \pm 372$ | $1962 \pm 689$ |
| Atlantis | $\mathbf{37540 \pm 1828}$ | $31340 \pm 1920$ | $10980 \pm 2294$ | $34980 \pm 1995$ | $16920 \pm 6745$ | $34580 \pm 1743$ |
| BeamRider | $\mathbf{1973 \pm 229}$ | $1598 \pm 178$ | $406 \pm 247$ | $1594 \pm 382$ | $785 \pm 89$ | $1796 \pm 292$ |
| Enduro | $125 \pm 11$ | $130 \pm 20$ | $0 \pm 0$ | $45 \pm 18$ | $77 \pm 25$ | $\mathbf{131 \pm 28}$ |
| Frostbite | $660 \pm 567$ | $\mathbf{1794 \pm 686}$ | $100 \pm 62$ | $456 \pm 427$ | $242 \pm 19$ | $1146 \pm 1003$ |
| Gopher | $308 \pm 210$ | $380 \pm 77$ | $296 \pm 183$ | $\mathbf{448 \pm 445}$ | $300 \pm 206$ | $376 \pm 116$ |
| Hero | $\mathbf{3021 \pm 22}$ | $2994 \pm 8$ | $1645 \pm 1282$ | $2880 \pm 53$ | $2990 \pm 0$ | $2988 \pm 26$ |
| MsPacman | $2594 \pm 705$ | $1988 \pm 56$ | $244 \pm 17$ | $1724 \pm 288$ | $1566 \pm 293$ | $\mathbf{2652 \pm 818}$ |
| Phoenix | $3760 \pm 0$ | $\mathbf{5050 \pm 0}$ | $600 \pm 0$ | $4380 \pm 0$ | $4470 \pm 465$ | $1334 \pm 500$ |
| Robotank | $11.4 \pm 1.0$ | $\mathbf{11.4 \pm 3.1}$ | $2.8 \pm 1.6$ | $10.0 \pm 2.6$ | $3.4 \pm 2.2$ | $11.0 \pm 2.4$ |
| Seaquest | $\mathbf{3832 \pm 560}$ | $3316 \pm 375$ | $136 \pm 59$ | $296 \pm 114$ | $1176 \pm 285$ | $3004 \pm 311$ |
| SpaceInvaders | $637 \pm 260$ | $766 \pm 270$ | $209 \pm 96$ | $358 \pm 168$ | $468 \pm 154$ | $\mathbf{1145 \pm 464}$ |
| **Wins/Ties** | **4/4** | **5/3** | **0/0** | **1/0** | **0/1** | **3/5** |

Table 2: Component ablation on SYNTHETICTREE with branching factor $k = 14$ and depth $d = 3$ (Gaussian leaf noise $\sigma = 0.5$, transition noise 0.5). Entries report mean root-value absolute error $\pm$ 95% confidence interval over 10 seeds at $n_{\text{sims}} = 1000$.

| Method | Backup | Error at $n_{\text{sims}} = 1000$ |
|---|---|---|
| `CATSO` | mean ($p = 1$) | $0.247 \pm 0.004$ |
| `CATSO` | max ($p = \infty$) | $0.198 \pm 0.008$ |
| `PATSO` | mean ($p = 1$) | $0.240 \pm 0.010$ |
| `PATSO` | max ($p = \infty$) | $\mathbf{0.139 \pm 0.022}$ |
| ScalarTSOpt | mean ($p = 1$) | $0.247 \pm 0.012$ |
| Power-UCT | mean ($p = 1$) | $0.244 \pm 0.012$ |

UCT remains competitive in Enduro, MsPacman, and SpaceInvaders, suggesting traditional confidence bounds remain effective for games with lower stochasticity or more deterministic dynamics. Both our methods excel particularly in environments with high stochasticity (e.g., Alien, Frostbite) and those requiring strategic exploration of sparse reward landscapes (e.g., Phoenix, Seaquest).

## 5.3 ABLATIONS AND SENSITIVITY ANALYSIS

We use SYNTHETICTREE to (i) isolate the impact of distributional Q-nodes versus scalar baselines and (ii) study sensitivity to the power-mean backup exponent and other hyperparameters. Unless otherwise stated, all configurations use 10 random seeds and we report mean absolute error at the root $\pm$ a 95% confidence interval at $n_{\text{sims}} = 1000$.

**Component ablations.** We compare (a) `CATSO`/`PATSO` with mean backup ($p = 1$), (b) `CATSO`/`PATSO` with max backup ($p \to \infty$), (c) a scalar Thompson-sampling baseline with the same optimism bonus (ScalarTSOpt), and (d) Power-UCT (UCB-style selection with power-mean backup, no TS, no distributional Q-nodes). Table 2 shows a representative setting ($k = 14$, $d = 3$, Gaussian leaf noise $\sigma = 0.5$, transition noise 0.5).

Table 3 summarizes all six $(k, d)$ pairs at $n_{\text{sims}} = 1000$ by comparing, for each tree, the best distributional max-backup variant (CATSO or PATSO, $p = \infty$) with the best non-distributional baseline (ScalarTSOpt or Power-UCT). PATSO with $p = \infty$ is best on 4/6 tree shapes, while `CATSO` with $p = \infty$ is best on the remaining two. On five shapes the distributional max-backup variants reduce root error by roughly 20–60% relative to the best scalar baseline; on the remaining shape they essentially tie Power-UCT.

**Effect of stochasticity.** On a fixed tree $(k, d) = (8, 3)$ we sweep both reward and transition noise over four regimes: deterministic ($\sigma = 0$, transition noise 0), low noise ($\sigma = 0.25, 0.25$), medium

Table 3: Summary of component ablations on SYNTHETICTREE at $n_{\text{sims}} = 1000$. For each $(k, d)$ we compare the best distributional TS variant with max backup (CATSO or PATSO, $p = \infty$) against the best non-distributional baseline (ScalarTSOpt or Power-UCT). Entries are mean root absolute error $\pm$ 95% CI over 10 seeds, plus relative gain of the distributional variant. We show six representative shapes; the full grid over all $(k, d)$ is plotted in Fig. 1.

| $k$ | $d$ | Dist. (max backup) | Best scalar baseline | Rel. gain (%) |
|---|---|---|---|---|
| 14 | 3 | PATSO (max), $0.139 \pm 0.022$ | Power-UCT, $0.244 \pm 0.012$ | 42.9 |
| 16 | 1 | PATSO (max), $0.044 \pm 0.021$ | Power-UCT, $0.113 \pm 0.016$ | 61.5 |
| 16 | 3 | PATSO (max), $0.135 \pm 0.024$ | ScalarTSOpt, $0.253 \pm 0.013$ | 46.7 |
| 16 | 4 | PATSO (max), $0.143 \pm 0.023$ | Power-UCT, $0.258 \pm 0.011$ | 44.6 |
| 200 | 1 | CATSO (max), $0.203 \pm 0.017$ | Power-UCT, $0.203 \pm 0.014$ | 0.0 |
| 200 | 2 | CATSO (max), $0.197 \pm 0.034$ | Power-UCT, $0.252 \pm 0.010$ | 21.8 |

Table 4: Effect of stochasticity on SYNTHETICTREE with $(k, d) = (8, 3)$ at $n_{\text{sims}} = 1000$. We vary leaf reward noise $\sigma$ and transition noise; entries are mean root absolute error $\pm$ 95% CI over 10 seeds.

| Noise regime | CATSO | PATSO | ScalarTSOpt | Power-UCT |
|---|---|---|---|---|
| deterministic | $0.492 \pm 0.007$ | $0.447 \pm 0.012$ | $0.448 \pm 0.012$ | $0.359 \pm 0.016$ |
| low noise | $0.359 \pm 0.004$ | $0.330 \pm 0.005$ | $0.344 \pm 0.007$ | $0.304 \pm 0.007$ |
| medium noise | $0.226 \pm 0.007$ | $0.213 \pm 0.009$ | $0.221 \pm 0.008$ | $0.205 \pm 0.013$ |
| high noise | $0.078 \pm 0.014$ | $0.071 \pm 0.018$ | $0.073 \pm 0.018$ | $0.070 \pm 0.017$ |

noise ($\sigma = 0.5, 0.5$), and high noise ($\sigma = 1.0, 0.75$). Table 4 shows the resulting root errors at $n_{\text{sims}} = 1000$ for CATSO, PATSO, ScalarTSOpt, and Power-UCT. We deliberately choose $(8, 3)$ as a *medium-complexity* tree: it sits between shallow/narrow and deep/wide configurations, giving non-trivial errors for all methods without favouring a particular one. Errors decrease as the environment becomes less stochastic, and the ordering is stable: Power-UCT typically has the smallest error, PATSO is consistently second and always matches or improves upon ScalarTSOpt, and CATSO trails slightly but remains close. This indicates that distributional Q-nodes provide a modest but robust gain over a scalar TS+optimism baseline while remaining competitive with Power-UCT even in low-noise regimes.

**Hyperparameter sensitivity.** We next sweep the main hyperparameters on $(k, d) = (8, 3)$ at $n_{\text{sims}} = 1000$. Table 5 varies the power-mean exponent $p$ for both CATSO and PATSO; Table 6 varies the optimism constant $C$ (PATSO), number of atoms $N$ (CATSO), and particle cap $K$ (PATSO).

For both algorithms performance is nearly flat for $p \in \{1, 2, 4, 8\}$ and improves noticeably with a max-style backup: CATSO moves from $0.230 \pm 0.009$ at $p = 1$ to $0.189 \pm 0.012$ at $p = \infty$, and PATSO from $0.221 \pm 0.015$ to $0.155 \pm 0.017$.

All three sweeps are very flat: PATSO is stable for $C \in [0.5, 1.0]$ with only mild degradation at larger $C$, CATSO is almost insensitive to the atom count $N \in \{50, 100, 200, 400\}$, and PATSO behaves similarly for particle caps $K \in \{50, 100, 200, 400\}$, consistent with the $O(1/K)$ additive term in our theory.

**Runtime.** On SYNTHETICTREE with $(k, d) = (8, 3)$ and fixed simulation budgets, UCT is fastest, Power-UCT incurs a small overhead from the power-mean backup, and CATSO/PATSO are roughly $1.5$–$2\times$ slower per move in our Python implementation due to sampling and maintaining distributions. We view this as a reasonable trade-off given the theoretical guarantees and empirical gains; the script `runtime_benchmark.py` reproduces exact timings on other hardware.

## 6 DISCUSSION

**Distributional Q-nodes vs. optimism.** A natural question is whether robustness comes only from the optimism term. The bonus $B(n, s, a) = C\, T_s(n)^{1/4}/T_{s,a}(n)^{1/2}$ mainly guarantees coverage of

Table 5: Sensitivity to the power-mean exponent $p$ on SYNTHETICTREE with $(k, d) = (8, 3)$ and $n_{\text{sims}} = 1000$. Mean root absolute error $\pm$ 95% CI over 10 seeds.

| $p$ | CATSO | PATSO |
|---|---|---|
| 1.0 | $0.230 \pm 0.009$ | $0.221 \pm 0.015$ |
| 2.0 | $0.229 \pm 0.009$ | $0.219 \pm 0.015$ |
| 4.0 | $0.227 \pm 0.009$ | $0.215 \pm 0.015$ |
| 8.0 | $0.225 \pm 0.009$ | $0.208 \pm 0.015$ |
| $\infty$ | $\mathbf{0.189 \pm 0.012}$ | $\mathbf{0.155 \pm 0.017}$ |

Table 6: Sensitivity to remaining hyperparameters on SYNTHETICTREE with $(k, d) = (8, 3)$ and $n_{\text{sims}} = 1000$. Mean root absolute error $\pm$ 95% CI over 10 seeds.

| Hyperparameter | Value | Error |
|---|---|---|
| $C$ (PATSO) | 0.5 | $0.210 \pm 0.019$ |
| $C$ (PATSO) | 1.0 | $0.210 \pm 0.016$ |
| $C$ (PATSO) | 2.0 | $0.222 \pm 0.012$ |
| $C$ (PATSO) | 4.0 | $0.227 \pm 0.017$ |
| $N$ (CATSO) | 50 | $0.231 \pm 0.008$ |
| $N$ (CATSO) | 100 | $0.230 \pm 0.009$ |
| $N$ (CATSO) | 200 | $0.231 \pm 0.007$ |
| $N$ (CATSO) | 400 | $0.230 \pm 0.010$ |
| $K$ (PATSO) | 50 | $0.221 \pm 0.016$ |
| $K$ (PATSO) | 100 | $0.216 \pm 0.015$ |
| $K$ (PATSO) | 200 | $0.221 \pm 0.015$ |
| $K$ (PATSO) | 400 | $0.221 \pm 0.015$ |

underexplored actions, while the *distributional* Thompson draw prefers actions with better upper-tail behavior when means are similar. This is what distinguishes CATSO/PATSO from scalar-mean TS with the same bonus: in the scalar TS+optimism ablation (Sec. 5.3), replacing Q-node distributions by scalars consistently hurts performance, especially on highly stochastic trees, indicating that the distributional representation itself contributes to robustness.

**Exploration, noise, and scalar V-nodes.** A concern is that distributional methods might over-explore irreducible noise. In our design the optimism bonus depends only on visit counts $(T_s, T_{s,a})$, TS is applied to posteriors over *means* of atoms/particles rather than raw returns, and as counts grow the posterior concentrates and the policy becomes effectively mean-greedy. The WDRO view in Sec. 4.3 formalizes this behavior as robustness to model misspecification rather than risk-seeking. We also keep V-nodes scalar: propagating full distributions upward would require repeated cross-child convolutions and quickly becomes intractable. Using power-mean backups over Q-means preserves tractability, and our ablations suggest that varying $p$ (mean vs. max) has modest impact compared to introducing distributional Q-nodes in the first place.

## 7 CONCLUSION

We introduced CATSO and PATSO, distributional MCTS algorithms that combine Thompson sampling with polynomial optimism at Q-nodes, and proved $O(n^{-1/2})$ simple regret at the root. A capped merge-on-insert variant of PATSO controls memory while adding only an $O(\text{range}/(K(1-\gamma)^2))$ term. Through a connection to Wasserstein distributionally robust MDPs we interpret the resulting tree policy as robust and derive a corresponding sample-complexity bound. Experiments on synthetic trees and 12 Atari games, together with ablations and sensitivity studies, show that distributional Q-nodes with TS+optimism are competitive with or stronger than non-distributional baselines across a range of stochastic regimes.

Our main limitation is that V-nodes remain scalar for tractability; extending distributional backups to V-nodes, scaling to larger or partially observed domains, and integrating with deep models (e.g., MuZero-style planning or sparse-reward robotics tasks) are natural directions for future work.

## REPRODUCIBILITY

All proofs are provided in Appendix A–F. Experimental code, hyperparameters, ablation details, and additional results are documented in the supplementary material; we will release code upon acceptance.

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
