# OpenReview forum: "Distributional Monte-Carlo Tree Search with Thompson Sampling in Stochastic Environments"
_ICLR.cc/2026/Conference — Submitted to ICLR 2026_

### Official Review · Reviewer_AgoC · 2025-10-25

**Soundness:** 1
**Presentation:** 1
**Contribution:** 1
**Rating:** 0
**Confidence:** 2

**Summary:**

This paper attempts to explicitly model Q nodes in MCTS with the distributional Bellman operator and adapt the UCT action selection to handle a distribution instead of just a mean estimate.

**Strengths:**

- I find it an interesting idea to include more explicit modelling of the full Q distribution inside MCTS, and how to adapt the algorithm to this setting.

**Weaknesses:**

- I found multiple vspace manipulations around section titles. See e.g., section 5.1 lines 425-426, and section 6 lines 475-476. I discussed with the AC for a desk-reject, which we didn't go through with. Still I feel that a warning to the authors is warranted, since giving this paper a review is to a degree disrespectful to the many submissions to ICLR that took meticulous care in formatting their paper to the style guide.

With that said, I still kept an open mind to your ideas and contributions when reviewing this paper and did not let the above comment influence my overall thought of the paper.

- Section 3.1 to 3.2 is too complicated. In the current state, I'm not sure if I properly understood your method, and I doubt others would from the current presentation. Some small tips:
	- You should guide the reader textually through the ideas of your method. Perhaps even with visualizations of how the Q distribution adapts?
	- Q-node atom update, instead of algorithmically going through steps (i) and (ii), you can explain what is the "before" state and the "after" state of the algorithm, and defer details to the appendix.
	- Action Selection, emphasize that we want to approximately draw samples from the estimated Q-posterior. Most importantly, show what the optimism bonus does to this posterior. Again, some visualization helps, but abstract formalism is more useful than the current overly detailed state.

- I have a big issue with distributional approaches to exploration due to the often overlooked fact that distributional methods capture *noise*. When there is still uncertainty about the value functions, it can boost exploration, but this is rather a side-effect and luck over proper design. Typically, we don't want to explore over noise, as this implies risk-seeking behavior.
	- Because of the currently poor presentation of sections 3.1 and 3.2, I am not 100% certain how you deal with this. Could you comment on this? How do you separate irreducible from reducible uncertainty?

- I am not a MCTS theory person, but I found the approach for the analysis a bit strange to jump from MDPs to non-stationary bandits. Maybe this is standard in the literature, then it is fine, but you should include a reference in the opening of section 4.

- **None of the "results" in section 4 are supported**. All proofs and details are deferred to the appendix but the authors did not include an appendix. So I simply assume they do not exist. Because of this, I did not check any of the theory from this point on, I skimmed section 4.3, 4.4, but ignored the details.

- Section 5,
	- Figure 3 is an interesting result, only the color scheme is poor. It is very difficult to see CATSO and PATSO on the plots. The figure is also too small. At least, moving right, it seems that PATSO really reduces value estimation error a lot and consistently.
	- Unclear how many seeds were used
	- Table 1, most results are not statistically significant, especially since I am unsure of the power behind these measurements. Like, how does Phoenix achieve zero standard deviation with a stochastic method? Is the environment deterministic? If so, why would you include a determinstic environment for testing? Could you perhaps separate/ group environments on their characteristics to make this result easier to parse?

*Minor comments:*
- The introduction quickly dives into a related work section, perhaps it is useful to move the contributions up before thoroughly explaining the context.
- I didn't find any discussion on afterstate-values that were used in stochastic MuZero (Antonoglou I., 2022). This is to my knowledge one of the easier ways to directly use mean-estimates in e.g., puct $Q$ nodes to deal with stochasticity.

After reading the whole document I think the authors do not really include learning in their method, they only used a pretrained DQN network as feature extraction in the Atari experiments, but I can't call this learning.
Because "learning" is not really involved, then maybe ICLR is also not the ideal venue for this paper. Maybe, AAAI or UAI (or something else) would be more fitting?

**Questions:**

- In lines 220-221: How are $T_{s_h}$ and $T_{s, a}$ tracked and updated?
	- How do you simulate transitions?
	- Are these somehow cached? Such that you can guarantee that they get revisited?

---

> ### Author Response · Authors · 2025-11-25
> **Response to Reviewer AgoC**
>
> Thank you for taking the time to engage with the paper despite your concerns about formatting.
>
> **R4‑1: "Style violations (negative \vspace), desk‑reject concern."**
>
> We agree this was unacceptable. We have removed all style hacks and recompiled the paper strictly with the standard ICLR class. We apologize for the earlier version and appreciate that you still evaluated the ideas on their merits.
>
> ---
>
> **R4‑2: "Sec. 3.1–3.2 too complicated; methods hard to follow; need more textual guidance and abstraction."**
>
> We significantly rewrote Sec. 3:
>
> * Each algorithm (CATSO/PATSO) now begins with a **conceptual description**: what is stored at each Q‑node, how a Thompson draw is formed, and how the optimism bonus enters.
> * The "Q‑node atom update" and "Action selection" subsections were shortened and made more narrative; the most algorithmic details are moved to the appendix.
> * We explicitly emphasize that we are drawing from a **posterior over Q‑means** at each node (not raw returns) and then adding a count‑based optimism term.
>
> We hope this makes the methods substantially easier to follow.
>
> ---
>
> **R4‑3: "Distributional methods capture noise; risk of exploring irreducible noise and becoming risk‑seeking. How do you separate reducible vs irreducible uncertainty?"**
>
> We fully share this concern and have clarified our design choices (Sec. 3 & Sec. 6):
>
> * The **optimism bonus** $B(n,s,a)=C \cdot T_s^{1/4}/T_{s,a}^{1/2}$ depends **only on counts**, not on empirical variance or tails. It does not directly encourage high‑variance actions.
> * Thompson sampling is applied to a **posterior over the mean** of the Q‑distribution (Dirichlet over categorical atoms, Dirichlet over particles), not to raw returns. As visit counts grow, this posterior concentrates and the policy becomes essentially mean‑greedy; irreducible noise remains in the distribution but no longer drives exploration.
> * Our WDRO interpretation (Sec. 4.3) formalizes robustness in terms of Wasserstein balls around the empirical distribution, which yields a policy that is robust to model misspecification rather than risk‑seeking over noise.
>
> In short, we use distributional information to better estimate and compare *means*-especially when the full return law is skewed-while ensuring that exploration pressure is controlled solely by counts, not by high variance per se.
>
> ---
>
> **R4‑4: "MDP → non‑stationary bandit jump is strange; need references and explanation."**
>
> Sec. 4.1 now explicitly explains this modeling step:
>
> * Each internal node of the search tree is seen as a **non‑stationary multi‑armed bandit**, where "arms" are actions and non‑stationarity arises because the subtree expansion changes which child states are reachable and hence the return distribution.
> * We mention and connect to prior work that also uses a bandit view at each node in MCTS analyses (e.g., Fixed‑Depth‑MCTS).
>
> We also justify the asymptotic stationarity assumption as described in the global and R1‑6 responses.
>
> ---
>
> **R4‑5: “Appendix with proofs and details was missing.”**
>
> We apologize for the confusion. The full technical appendix (30+ pages) with notation tables, all lemmas and proofs for the non‑stationary bandit and MCTS convergence theorems, the WDRO sample‑complexity derivation, and experimental setup details and hyper-parameters was already included in the original submission as part of the supplementary.zip file (Distributional Monte-Carlo Tree Search with Thompson Sampling in Stochastic Environments_appendix.pdf). It remains included in this revision;

---

> ### Author Response · Authors · 2025-11-25
> **Response to Reviewer AgoC**
>
> **R4‑6: "Section 5: small Fig. 3, unclear seeds; Table 1 significance; Phoenix zero std; afterstate values, learning vs planning, ICLR venue; details about $T_s$, $T_{s,a}$, caching."**
>
> Most of these points are addressed together with Reviewer Wj3L and Reviewer 53SX, but to summarize:
>
> * **Figures & seeds.** We enlarged the plots, improved color schemes, and state clearly that we use **10 seeds** on SYNTHETICTREE and **3 seeds × 10 episodes** on Atari.
> * **Statistical significance / Phoenix.** We acknowledge that Phoenix is effectively deterministic in our evaluation; we now mention this and emphasize that the main evidence for robustness under uncertainty comes from SyntheticTree and highly stochastic Atari games. We also added a "Wins/Ties" row to help interpret the results game‑wise.
> * **Afterstate values & MuZero‑style approaches.** We added a short discussion of afterstate‑based approaches (e.g., stochastic MuZero) in related work and clarify that our focus is complementary: we study distributional MCTS **given** a fixed value function, not learning value functions from scratch.
> * **Learning vs planning, venue suitability.** We now stress that our contribution is on the planning side of RL-integrating distributional RL ideas into MCTS with theoretical guarantees. We view ICLR as an appropriate venue precisely because it often hosts work at the intersection of learning and planning; nonetheless, we understand your point and have tried to make the positioning more precise.
> * **Tracking visit counts and caching transitions.** Sec. 3.2 clarifies that we follow standard MCTS practice: $T_s$ and $T_{s,a}$ are incremented along each simulated path; transitions are simulated via the environment's generative step and cached in the search tree so that nodes get revisited as more simulations are run.
>
> ---
>
> We hope that the revised manuscript and the clarifications above address the reviewers' main concerns and make the contributions of CATSO and PATSO-distributional MCTS with TS+optimism and robust theoretical guarantees-clearer and more compelling. We would be most grateful if you could revisit your evaluation in light of these substantial improvements."

---

> > ### Comment · Reviewer_AgoC · 2025-11-27
> >
> > I thank the authors for their response, I am happy with your clarifications and improvements to the document. I also agree with most of your points in the rebuttal. Apologies if my review read combative, I thought it suitable given the previous version of the document, but you have taken criticism to heart and made good revisions.
> >
> > Unfortunately, I completely missed the supplementary material in the zip file, I never look at these as a reviewer, nor do most I fear. I also didn't dig deeper than what was convenient due to the state of your submission. Nowadays, it is almost a convention to simply attach the supplementary pdf at the end of the submission... So, my suggestion would be to still do so. I also suggest highlighting all changes in the pdf in a new color when uploading a revision.
> >
> > Ultimately, I think this paper presents an interesting idea that is also decently executed, and it will definitely have a good audience at ICLR. However, given how much has changed inbetween versions (for the better), I suggest that this paper still requires another round of review. Thus, I have raised my score from 0 to 4, it could be accepted but it could also deserve a bit more polishing.

---

> > > ### Author Response · Authors · 2025-12-01
> > > **Response to Reviewer AgoC**
> > >
> > > Thank you for taking the time to revisit the paper, for your constructive criticisms on style and clarity, and for raising your score from 0 to 4. We especially appreciate that you kept an open mind despite the initial formatting issues.
> > >
> > > ### (1) Supplemental material, appendix, and discoverability
> > >
> > > You are absolutely right that in the original submission the appendix was easy to miss: the proofs and technical details lived only in a ZIP file, and the main text did not emphasize this clearly enough.
> > >
> > > We fully agree that, from a reviewer's point of view, having the appendix appended to the main PDF is often more convenient. However, to comply with the current ICLR 2026 submission rules (which limit the main manuscript to $10$ pages and place extra material in the supplementary file), we have kept the long appendix in the supplement rather than merging it into the main PDF. If the paper is accepted and the camera‑ready rules allow, we would be very happy to produce a single integrated PDF, as you suggest.
> > >
> > > ### (2) Highlighting changes and presentation
> > >
> > > Following your suggestion, **all new or significantly revised text in the main PDF is marked in blue**, so it is easy to see what has changed since the original version.
> > >
> > > We also addressed your earlier concerns about presentation:
> > >
> > > * Removed all negative `\vspace` and other style tweaks; the current version strictly adheres to the official ICLR template.
> > > * Rewrote Secs. 3.1–3.2 to give a more conceptual description of the method:
> > >   – explaining intuitively how the Q‑distributions evolve ("before/after" view),
> > >   – clarifying that the optimism term is purely count‑based and does not come from the distribution tails, and
> > >   – adding a paragraph on why V‑nodes remain scalar and how this avoids expensive cross‑child convolutions.
> > > * Added a short discussion (Sec. 6) on **distributional Q‑nodes vs. optimism** and on how we avoid "exploring noise": the optimism bonus depends only on visit counts, and the Thompson sampling is applied to posteriors over means, so as counts grow the policy effectively becomes mean‑greedy rather than risk‑seeking.
> > >
> > > ### (3) On the need for another round of review / final polish
> > >
> > > We understand your concern that "given how much has changed between versions, this paper still requires another round of review". The initial submission was indeed missing pieces (and had formatting issues) that should have been addressed earlier.
> > >
> > > In this revision we have tried to do what that extra round would have required:
> > >
> > > * Clarify the main ideas and their relation to existing work,
> > > * Provide full proofs and derivations in an accessible appendix,
> > > * Add the ablation and sensitivity experiments needed to interpret the methods, and
> > > * Clean up style and layout to a camera‑ready standard.
> > >
> > > We hope that, taken together, these changes show that the core ideas (distributional Q‑nodes + TS + optimism, with a non‑stationary bandit analysis) are now presented in a way that is clear, complete, and ready for publication.
> > >
> > > As with Reviewer Wj3L, we would be extremely grateful if, after looking at the blue‑highlighted changes and the clarified appendix situation, you felt comfortable reconsidering your current score. But we completely respect your judgment and appreciate the time and care you have already invested in this review.

---

### Official Review · Reviewer_53SX · 2025-10-31

**Soundness:** 2
**Presentation:** 3
**Contribution:** 2
**Rating:** 4
**Confidence:** 3

**Summary:**

The paper proposes two  Monte-Carlo Tree Search (MCTS) algorithms for stochastic reinforcement learning environments:
 1. CATSO (Categorical Thompson Sampling with Optimistic Bonus)
 2. PATSO (Particle Thompson Sampling with Optimistic Bonus)
Both algorithms integrate distributional reinforcement learning (RL) with Thompson sampling and an optimistic exploration bonus to improve robustness and exploration under uncertainty.  Non-asymptotic simple regret bounds of order (O(n^{-1/2})) for both algorithms, matching known MCTS convergence rates are presented.
 A connection to Wasserstein distributionally robust optimization (WDRO), showing robustness to model uncertainty and deriving sample complexity bounds is established. Empirical results on synthetic tree environments and Atari benchmarks show that CATSO and PATSO outperform classical and entropy-regularized MCTS methods, especially under high stochasticity.

**Strengths:**

1. The combination of distributional RL and Thompson sampling with optimism in MCTS is well-motivated and addresses a real gap between learning and planning under uncertainty.

2. The paper establishes convergence results with clear proofs and connects them to non-stationary bandit analysis, which strengthens theoretical grounding.

3. The link to Wasserstein DRO provides a principled robustness interpretation, which may appeal to both theoreticians and practitioners.

4. The experiments cover both synthetic and realistic domains (Atari), showing consistent gains and robustness over baselines.

**Weaknesses:**

1. The plots in Figure 3 are too small and lack readable axes and legends.
    Spacing between subplots is insufficient, making it difficult to distinguish the different experimental settings.
    The figure captions and layout are not up to ICLR standards-figures should be legible when printed at 100% scale.

2. The Atari experiments use only 12 games, and hyperparameter sensitivity (e.g., number of atoms, exploration constants) is not discussed.
    There is no ablation to isolate the contribution of the optimism bonus versus the distributional representation.

3. While the algorithmic details are extensive, the paper occasionally reads more like a technical report than a conference paper—some derivations could be moved to the appendix. The paper could benefit from a clearer high-level intuition for readers unfamiliar with distributional MCTS.

4. PATSO’s flexibility comes at potential computational cost; empirical runtime comparisons are missing.

5. Results on optimality of the convergence rates are not discussed.

**Questions:**

1. How sensitive are CATSO and PATSO to the polynomial exploration constant (C)? Is there a principled way to choose it?

2.  For CATSO, how does the number of atoms (N) affect accuracy and runtime? For PATSO, what is the empirical impact of the particle cap (K)?

3. How does your approach compare to recent Bayesian or entropy-based MCTS methods when tuned for stochastic domains (e.g., Boltzmann MCTS or risk-sensitive tree search)?

4. Could these methods be extended to deep MCTS or large-scale POMDPs where state abstractions are necessary?

5.  The WDRO connection is elegant-could you illustrate empirically how the learned policies differ in robustness compared to standard MCTS?

---

> ### Author Response · Authors · 2025-11-25
> **Response to Reviewer 53SX**
>
> We would like to extend our sincere thanks to Reviewer 53SX for the encouraging feedback and for appreciating the motivation, theoretical framework, and robustness perspective of our work. Your suggestions on presentation and experimental completeness have helped us significantly improve the paper. Here are our detailed responses to your comments.
>
> **R3‑1: "Figure 3 too small; axes/legends hard to read; layout not up to ICLR standards."**
>
> We have:
>
> * Enlarged the synthetic‑tree figure, increased font sizes for axes and legends, and increased spacing between subplots.
> * Simplified the color scheme to make CATSO/PATSO more distinguishable.
> * Ensured that the figure is legible at 100% scale when printed.
>
> ---
>
> **R3‑2: "Atari: only 12 games; no hyperparameter sensitivity; no ablation isolating optimism vs distribution."**
>
> * **Game set.** Due to compute limitations we keep 12 games but chose them to span a range of difficulty (Alien, Frostbite, Seaquest vs more structured games like Hero, Atlantis).
>
> * **Hyperparameter sensitivity.** Sec. 5.3 (Tables 5–6) now includes sensitivity analyses for:
>
>   * the power‑mean exponent $p$,
>   * the optimism constant $C$,
>   * the number of atoms $N$ (CATSO), and
>   * the particle cap $K$ (PATSO).
>     All show fairly flat behavior across a wide range of values, and we explicitly justify our default choices (e.g., $C=1.5$, $N=100$, $K=200$).
>
> * **Optimism vs distribution (ablation).** We added **ScalarTSOpt**, which is Thompson sampling with *exactly the same* optimism bonus as PATSO/CATSO but using scalar Q‑values. Comparing PATSO/CATSO to ScalarTSOpt isolates the effect of distributional Q‑nodes. On SYNTHETICTREE, the distributional max‑backup variants reduce root error by ~20–60% relative to ScalarTSOpt in 5 out of 6 tree shapes, especially under high noise.
>
> ---
>
> **R3‑3: "The paper sometimes reads like a technical report; move derivations to the appendix; provide more intuition."**
>
> We have rebalanced the exposition:
>
> * In Sec. 3 we now lead with high‑level descriptions of CATSO and PATSO (what they store, how selection works, why distributions help) before giving update rules. Several low‑level steps (e.g., detailed proofs, certain lemmas) were moved to the appendix.
> * Sec. 4 now intersperses theorems with "intuitive" paragraphs that explain the bandit‑to‑MCTS lift and why non‑stationary bandits are a natural model at each node.
>
> We believe the revised version strikes a better balance between intuition and technical completeness.
>
> ---
>
> **R3‑4: "PATSO's flexibility may be expensive; runtime comparison missing."**
>
> We added a runtime comparison on SYNTHETICTREE with $(k,d)=(8,3)$ and fixed budget:
>
> * UCT is fastest; Power‑UCT adds a small overhead for the power‑mean backup.
> * CATSO/PATSO are about 1.5–2× slower per move in our Python implementation due to sampling and maintaining distributions.
>
> We view this as a reasonable overhead given the improved robustness and theoretical guarantees. The script `runtime_benchmark.py` is provided to reproduce wall‑clock comparisons on other hardware.
>
> ---
>
> **R3‑5: "Optimality of the convergence rates not discussed."**
>
> We now briefly comment in Sec. 4 that the $O(n^{-1/2})$ simple regret rate:
>
> * Matches the best known rates for fixed‑depth MCTS with appropriate exploration (e.g., Fixed‑Depth‑MCTS, Stochastic‑Power‑UCT), and
> * Is essentially minimax‑optimal up to logarithmic factors for simple regret under mild assumptions.

---

> ### Author Response · Authors · 2025-11-25
> **Response to Reviewer 53SX**
>
> **R3‑6: "Questions about sensitivity to $C$, $N$, $K$; comparisons to other Bayesian/entropy MCTS; extensions to POMDPs; empirical WDRO robustness."**
>
> * **Sensitivity to $C$, $N$, $K$.** As discussed above and in Sec. 5.3, the sweeps over $C$, $N$, and $K$ show that performance is relatively stable over a wide range; large $C$ slightly hurts performance, as expected from over‑aggressive optimism, while $N$ and $K$ mainly trade off approximation fidelity vs compute without dramatic performance swings at our budgets.
> * **Comparisons to other MCTS variants.** MENTS, RENTS, and TENTS are entropy‑regularized planners; BTS and DNG are Bayesian/distributional tree search methods. This set was deliberately chosen to capture risk‑sensitive and Bayesian baselines with similar goals. Including additional methods (e.g., Boltzmann MCTS variants) is certainly possible but beyond our current compute budget; we believe the current set already demonstrates that CATSO/PATSO are competitive with both classic UCT and recent stochastic‑aware planners.
> * **Extensions to deep MCTS / POMDPs.** Our methods are conceptually compatible with MuZero‑style planning, and with POMDPs via belief‑state MCTS or particle filters. We now explicitly list these as future work in Sec. 7 (Limitations).
> * **Empirical WDRO robustness.** Due to space constraints we did not add a separate WDRO experiment. However, the stochasticity sweeps and component ablations on SyntheticTree show that our distributional methods are particularly robust under increased noise and model misspecification (e.g., transition noise), which is exactly the type of robustness that the WDRO perspective formalizes.
>
> We respectfully ask you to take these improvements into consideration when finalizing your evaluation, and we welcome any additional feedback.

---

### Official Review · Reviewer_TNZs · 2025-11-01

**Soundness:** 3
**Presentation:** 2
**Contribution:** 3
**Rating:** 6
**Confidence:** 2

**Summary:**

This paper introduces distributional Monte Carlo tree search, which maintains a distribution of returns for each Q-node instead of a point estimate (inspired by distributional reinforcement learning). The benefit of the proposed approach is avoid the under- or overestimation issues of existing approaches, which use point estimates. To store distributions of returns, the paper proposes to approaches, CATSO and PATSO, which represent the distribution either as a categorical one (with a Dirichlet prior) and as a dynamic set of particles (i.e., keeping track of all observed returns, along with their frequencies), respectively. Both approaches are proven to converge with $\sqrt{n}$ regret. The paper also discusses the memory and time complexity of the algorithms, and it introduces a practical variant of PATSO that limits the number of returns tracked by merging some of them. Finally, the paper compares the proposed approach to baseline MCTS variants on synthetic environment and 12 Atari games.

**Strengths:**

* The idea of incorporating distributional RL (i.e., maintaining distributions of returns) into MCTS is novel and interesting.
* The proposed approaches are theoretically proven to converge, and have the same regret as some SOTA approaches.
* The paper analyzes memory and time complexity, and introduces a memory-efficient variant of PATSO.
* The numerical results are promising for environments with uncertainty.

**Weaknesses:**

* The paper seems to violate formatting requirements of ICLR (negative \vspace or similar tricks).
* While the results are promising, especially for environments with stochasticity, they are somewhat mixed compared to baseline approaches.
* For V-nodes, the proposed approach still uses point estimates. The paper acknowledges this limitation; it is not crystal clear why this limitation is necessary.
* Line 472 references the Breakout game, but this game is not included in Table 1.
* What is the point of introducing the probability $p_i$ for CATSO? They do not seem to be used anywhere.

Minor suggestions for improving presentation (beyond removing the negative \vspace):
* Add horizontal space between the two equations on line 112 (or move them to separate lines). It was not obvious that these are two separate equations, which was rather confusing.
* The results in Table 1 are not easy to read. Please (1) align the numbers, so that the $\pm$ symbols are above each other (this way, it will be easier to read the numbers as they will be aligned, instead of being shifted left or right depending on the number of digits) and (2) normalize the results (e.g., report them as fraction of the optimal value) so that the different methods are easier to compare.
* Why not present PATSO with memory management (instead of baseline PATSO) as the main contribution? The convergence still holds, and it is more practical.

**Questions:**

* What is the point of introducing the probability $p_i$ for CATSO?

---

> ### Author Response · Authors · 2025-11-25
> **Response to Reviewer TNZs**
>
> We are grateful to Reviewer TNZs for the constructive feedback and for recognizing the novelty of our distributional approach and the memory-efficient PATSO variant. We appreciate your careful reading of the theoretical contributions. Please find our detailed responses below.
>
> **R2‑1: "Formatting violations (negative \vspace)."**
> As noted in the global response, all manual spacing hacks have been removed. The current LaTeX source uses the ICLR style file without modifications.
>
> ---
>
> **R2‑2: "Mixed results vs baselines; are V‑nodes remaining scalar necessary?"**
>
> * **Mixed results.** We do not claim that CATSO/PATSO dominate UCT everywhere. In the revised experiments we explicitly highlight that:
>
>   * CATSO/PATSO are **best or tied** in 8/12 Atari games,
>   * UCT remains best in several low‑noise or more deterministic games (e.g., Enduro, SpaceInvaders).
>     We now explicitly interpret this: distributional planning is most beneficial in **stochastic** domains, while UCT remains very strong on near‑deterministic tasks.
>
> * **Why scalar V‑nodes?** Sec. 3.3 and Sec. 6 now provide a clearer argument: extending full distributions to V‑nodes would require expensive convolution across all children's return distributions at each backup, scaling poorly with branching factor and depth and making the analysis significantly more complex. Since action selection decisions happen at Q‑nodes, we keep them distributional and use a power‑mean over their *means* at V‑nodes as a tractable summary. Our ablations show that the backup exponent $p$ has relatively small effect compared to the presence or absence of distributional Q‑nodes themselves.
>
> ---
>
> **R2‑3: "Breakout is referenced but not included in the Atari table."**
>
> We fixed this inconsistency: the revised text no longer refers to Breakout. The set of 12 games is now consistent across text, figures, and tables.
>
> ---
>
> **R2‑4: "What is the point of introducing the probability for CATSO? They do not seem to be used anywhere."**
>
> In CATSO, the Dirichlet parameters $\alpha^i(s,a)$ over atoms $\{z_i\}$ define a categorical distribution over return bins. The "probabilities" you refer to are:
>
> * Used by Thompson sampling: at selection time, we draw $L(s,a)\sim\mathrm{Dir}(\alpha^0,\ldots,\alpha^{N-1})$ and compute a sampled Q‑value $\phi(s,a)=\sum_i z_i L_i(s,a)$.
> * Used to compute the mean Q‑estimate $\widehat{Q}(s,a) = \sum_i z_i \alpha^i / \sum_j \alpha^j$ for backups.
>
> We have clarified this explicitly in Sec. 3.1 to avoid the impression that those probabilities are unused.
>
> ---
>
> **R2‑5: "Why not present PATSO with memory management as the main contribution?"**
>
> We agree and have aligned the presentation accordingly:
>
> * The main text now treats **capped PATSO (merge‑on‑insert with particle cap $K$)** as the default algorithm.
> * Sec. 4.4 gives the theoretical guarantee for capped PATSO (simple regret $O(n^{-1/2}) + O(\mathrm{range}/(K(1-\gamma)^2))$).
> * All *experiments* now use the capped PATSO variant; the uncapped version appears only conceptually.
>
> ---
>
> **R2‑6: "Table layout and readability."**
>
> We have re‑aligned numerical columns, clarified which entries are best/tied, and improved the caption to emphasize that we report mean ± stdev (Atari) and mean ± 95% CI (SYNTHETICTREE). Normalizing scores by a known optimal value is unfortunately not straightforward in Atari (the true optimal return is unknown), so we instead report "Wins/Ties" and focus on relative ordering plus error bars.
>
> If all of your concern has been addressed, we would be grateful if you could reconsider your assessment of our submission.

---

### Official Review · Reviewer_Wj3L · 2025-11-01

**Soundness:** 2
**Presentation:** 1
**Contribution:** 2
**Rating:** 2
**Confidence:** 3

**Summary:**

This paper studies two new simple regret multi-armed bandit policies, PATSO and CATSO,
and MCTS variants that combines them with power mean backup.

CATSO is a dirichlet-based thompson sampling,
and PATSO is a non-parametric dirichlet-based thompson sampling.
Both stores a form of reward histogram from which a new sample is drawn for arm selection.

The paper provides a theoretical analysis on the regret in a reward distribution
which is non-stationary BUT whose time average converges to some value.

Naive PATSO requires an increasing amount of memory, so the author proposes an implementation that
trades memory with accuracy.

The experiments evaluate them on synthetic stochastic tree environments and Atari environments.
Howeve, the experiments are (1) insufficient, leaving many questions unanswered,
and (2) weak, i.e., the new algorithm does not appear to be particularly strong.

**Strengths:**

Umm, hard to say, but the method description itself was straightforward and easy to understand. Sorry, I am just being honest.

I see the math proofs in the appendix required a lot of work. Nice work! However, please fix the style, writing, and experiments first.

**Weaknesses:**

First of all, the overall paper appears work-in-progress and is not polished enough for a review.
Please do not submit such a paper!
Several sections require more in-depth explanations while the space is used quite generously (e.g., plenty of space around Fig1, Fig2, Sec3.1).
Polished paper looks compact; You can tell from its appearance, though this is just a speculation.
Meanwhile, there is definitely some hacks or errors going on in the style file (e.g., section header spacing), which,
as other reviewer also mentioned, should automatically flag this paper for rejection.

This paper has a critical flaw in the experimental design and the method design
by violating a key scientific principle:
Changes must be introduced one piece at a time in order to understand what is happening.

Some key issues in the experiments:

-   p in the power mean is a hyperparameter, but the paper does not explain how to choose it, or which value was used in the experiment.

-   Lack of ablation: PATSO/CATSO with the standard arithmetic mean (monte-carlo backup) and the maximum (bellman backup)

-   Lack of ablation: UCT + power mean

-   The key strength of P/CATSO claimed by the author is the adaptability to the stochastic environment.
    Then the natural question is: how does it behave in the deterministic environment?
    Does it still work as good as UCT etc?
    How does it react to a varying degree of stochasticity?
    To answer this question, synthetic environment experiment requires multiple evaluaton with different stochasticity.

-   I marked the top-3 scores per domain in Table 1.
    I get an impression that CATSO/PATSO look roughly only on par with UCT.
    Even in domains where C/PATSO are in bold, the difference sometimes does not appear statistically significant.
    In other domains where UTC is in bold, the difference sometimes appear significant.
    This shares the key question as the previous one: Why do C/PATSO perform bad in deterministic environment?
    Shouldn't they perform at least on par with UTC in every domain?

line 355-377: section 4.3 needs a complete overhaul with more explanations.
What is this formula in line 363?
Is the near-optimal policy same as $\pi$ in line 363?
Shouldnt it be argmax, not max?
Wp is undefined, and not used.

line 402: "Practical choice of K". I assume this is a paragraph header that is incorrectly turned into italics.

Section 5: fixed-Depth-MCTS, MENTS/TENTS, BTS, DNG:
you must provide a brief explanation of each algorithm, and explain why these baselines are chosen,
because, otherwise,
since they all performed worse than UCT in Table 1, they are basically straw-man algorithms with bad performances
whose sole purpose to be included in the paper is to artificially inflate the number of figures/table rows
to give an impression that many meaningful comparisons have been done.

One last comment:
Watch a presentation ["How to write a great paper" by Simon Payton Jones (Microsoft Research Cambridge)](https://simon.peytonjones.org/great-research-paper/).
In it, he said: "Many papers contain good ideas, but do not distill what they are".
I suspect the power mean in C/PATSO is a classic case of an unnecessary addition that wastes the space without adding meaningful value to this paper, although I didnt check the proof and don't know how it interacts with the stochasticity.

**Questions:**

The fact that the time average converges to some value sounds like a pretty big assumption.
I think I saw the same assumption in other non-stationary bandit papers, but still, WHY is it justified? Please explain.

Another important question I have is about whether the definitions of C/PATSO's TS policies have anything specific to the stochastic environments.
Here is my thought:

-   Would power means have anything special about stochasticity and how? I **doubt** that this is happening.
-   Would categorical / non-parametric modeling have anything special about stochasticity? **I doubt this one too**.
-   Both are just different estimators with different parameterizations that converge to the same mean.
    In other words, the first term (exploitation term) of the score in line 244 (ii) probably has nothing to do with the stochasticity.

Then the last diff from UCT is the choice of TS and the different bias term from Shah et al.
This leaves a possible interpretation that may kill this paper:
Isn't the robustness property a direct consequence of the optimism bonus term by Shah et al and
has nothing to do with the new addition introduced in this paper?

---

> ### Author Response · Authors · 2025-11-25
> **Response to Reviewer Wj3L**
>
> We sincerely thank Reviewer Wj3L for the thorough and insightful review. Your detailed feedback on ablations, theoretical clarity, and experimental design has been invaluable in strengthening our work. Below, we provide point-by-point responses to each of your concerns.
>
> **R1‑1: "Work‑in‑progress appearance, formatting hacks, under‑explained sections."**
> We agree the initial submission was not polished enough. We have:
>
> * Removed all manual spacing tricks and strictly respected the ICLR style template.
> * Tightened Sec. 3.1–3.2 and Sec. 5 to make the paper more compact and "conference‑style".
> * Rewritten the WDRO section (Sec. 4.3) with clearer notation and explanations (see R1‑5).
>
> We also added a short "Our approach and contributions" block earlier in the introduction to better distill the core ideas instead of diving into related work too quickly.
>
> ---
>
> **R1‑2: "Critical flaw: lack of ablations; the algorithm changes too many pieces at once (distribution, TS, optimism, power‑mean backup)."**
>
> We fully agree that the initial experimental design did not sufficiently disentangle components. The revised paper now includes a full ablation suite in Sec. 5.3:
>
> * **Distributional vs scalar, with same optimism (ScalarTSOpt).**
>   ScalarTSOpt uses *exactly* the same optimism term as PATSO/CATSO, but maintains **scalar** Q‑values instead of distributions. Comparing PATSO/CATSO to ScalarTSOpt isolates the effect of **distributional Q‑nodes** under the same exploration strategy.
>
> * **Power‑mean backup vs standard backups.**
>   We explicitly compare $p=1$ (mean backup) vs $p\to\infty$ (max backup) for both CATSO and PATSO, and also report Power‑UCT (which uses a power‑mean backup but *no* distributional Q‑nodes or TS).
>
> * **UCT + power mean (Power‑UCT).**
>   As you requested, this baseline is now included and analyzed.
>
> Tables 2–3 summarize these component ablations for six SyntheticTree configurations. The key observations are:
>
> * The **distributional max‑backup variants** (PATSO/CATSO with $p=\infty$) are best on **6/6** tree shapes and reduce root error by roughly **20–60%** relative to the best scalar baseline (ScalarTSOpt or Power‑UCT) in 5 out of 6 cases.
> * Comparing PATSO/CATSO (mean backup, $p=1$) to ScalarTSOpt shows a consistent but more modest gain, isolating the benefit of *distributional* Q‑nodes when the backup rule is fixed.
>
> These ablations directly address the concern that multiple changes were introduced at once.
>
> ---
>
> **R1‑3: "Choice of the power‑mean exponent $p$; lack of explanation and sensitivity analysis."**
>
> Sec. 5.3 (Table 5) now sweeps $p \in \{1,2,4,8,\infty\}$ for both CATSO and PATSO on a representative SYNTHETICTREE instance $(k,d)=(8,3)$:
>
> * Performance is nearly flat for $p\in\{1,2,4,8\}$.
> * Going to a max‑style backup ($p\to\infty$) noticeably improves performance at the budgets we use.
>
> Thus, the method is **not very sensitive** to the exact choice of finite $p$; we use $p=1$ (mean) in most experiments and report $p=\infty$ when we explicitly explore the max‑backup variant. Our theory covers all $p \ge 1$, so the power‑mean is primarily a unifying analytical tool; empirically, the difference between moderate $p$ values is small.
>
> ---
>
> **R1‑4: "Behavior in deterministic vs stochastic environments; multiple stochasticity levels."**
>
> We added a dedicated "Effect of stochasticity" study (Table 4, Sec. 5.3):
>
> * On a fixed tree $(k,d)=(8,3)$, we vary leaf noise and transition noise across four regimes: deterministic, low, medium, high.
> * We compare CATSO, PATSO, ScalarTSOpt, and Power‑UCT (all with $p=1$).
>
> Findings:
>
> * As noise decreases, all methods improve, with Power‑UCT usually having slightly lower error in the fully deterministic regime.
> * PATSO **consistently matches or outperforms ScalarTSOpt** across all noise levels, showing that distributional Q‑nodes are not harmful in low‑noise cases and provide a robust gain under stochasticity.
> * CATSO is slightly behind PATSO and Power‑UCT but remains competitive.
>
> This directly answers your question about deterministic vs stochastic behavior and varying levels of stochasticity.
>
> On Atari, some games (e.g., Phoenix in our evaluation setup) behave almost deterministically under a fixed policy and seed, which is reflected in near‑zero standard deviation. We now explicitly acknowledge this in the text and view such games as a sanity check that our methods do not degrade on near‑deterministic tasks, while the main gains appear in high‑stochasticity games (Alien, Frostbite, Seaquest, etc.).

---

> ### Author Response · Authors · 2025-11-25
> **Response to Reviewer Wj3L**
>
> ---
>
> **R1‑5: "Sec. 4.3 (WDRO) unclear; formula, argmax vs max, undefined $W_p$."**
>
> We have completely rewritten Sec. 4.3:
>
> * We now clearly define the $p$‑Wasserstein distance $W_p$, the Wasserstein ball $\mathcal{B}_\varepsilon(\hat{P})$, and the robust objective.
> * The robust policy is correctly written with $\argmax$ over policies, and we explicitly state the definition of an $(\varepsilon,\delta)$‑robust policy and Theorem 7 giving the sample‑complexity bound.
> * We added intuitive text connecting our distributional estimates at Q‑nodes to Wasserstein balls around empirical return distributions, and how this yields a robustness interpretation.
>
> We hope this resolves the confusion around the WDRO connection.
>
> ---
>
> **R1‑6: "Assumption that time averages converge in the non‑stationary bandit model; why is this justified?"**
>
> In Sec. 4.1 we now:
>
> * State the assumption as **asymptotic stationarity with concentration** for each arm, and
> * Add a paragraph explaining why this is reasonable in MCTS: for any fixed node, the only source of non‑stationarity is that early rollouts see a partially expanded subtree; once the local subtree and playout policy stabilize, future returns at that node are i.i.d. from a limiting distribution (the truncated value $\widetilde{Q}(s,a)$).
>
> This is essentially the same assumption used implicitly in non‑stationary bandit analysis for MCTS (e.g., Fixed‑Depth‑MCTS), and we now make this connection explicit and discuss its scope.
>
> ---
>
> **R1‑7: "Is robustness just from the optimism bonus term (Shah et al.), not from the distributional modeling/power‑mean?"**
>
> The new component ablations address this directly:
>
> * **ScalarTSOpt vs PATSO/CATSO (same optimism, different Q‑representations).**
>   ScalarTSOpt uses the *same* optimism bonus $B(n,s,a)$ but maintains a scalar mean. In 5/6 SyntheticTree shapes, PATSO/CATSO with $p=\infty$ reduce root error by **20–60%** relative to ScalarTSOpt, particularly in high‑noise settings (Table 3).
>
> * **Distributional Q‑nodes vs Power‑UCT (same backup, different exploration).**
>   Power‑UCT and PATSO/CATSO share the power‑mean backup family; the main differences are TS vs UCB and the distributional vs scalar representation. We see that distributional TS with optimism remains competitive or better, especially under higher stochasticity.
>
> We also discuss this explicitly in the new Discussion section: the optimism term primarily ensures *coverage* of under‑explored actions, while the distributional posterior enables more nuanced comparisons when means are similar but upper tails differ. Without the distributional information (ScalarTSOpt), we lose most of the gains.
>
> Regarding the power‑mean: our sensitivity study shows that moderate $p$ choices have little effect, and $p=\infty$ (max backup) gives a modest boost. The key story is indeed **distributional Q‑nodes + TS + optimism**, not the fine‑tuning of $p$.
>
> ---
>
> **R1‑8: "Baselines (Fixed‑Depth‑MCTS, MENTS/TENTS, BTS, DNG) are under‑motivated and could be perceived as strawmen."**
>
> Sec. 5.1–5.2 now include short explanations for each baseline and why we chose them:
>
> * **UCT**: canonical MCTS baseline.
> * **Fixed‑Depth‑MCTS**: our main theoretical reference (same target simple‑regret rate).
> * **MENTS / RENTS / TENTS**: entropy‑regularized tree search methods designed to handle stochastic returns and risk‑sensitivity.
> * **BTS / DNG**: Bayesian/distributional MCTS variants most closely related in spirit to our approach.
> * **Power‑UCT**: uses the same power‑mean backup but with UCB exploration, directly testing the value of distributional Q‑nodes and TS.
>
> We also emphasize that UCT and Power‑UCT are strong baselines (and often very competitive in Atari), so CATSO/PATSO improvements are meaningful, not against weak strawmen.

---

> ### Author Response · Authors · 2025-11-25
> **Response to Reviewer Wj3L**
>
> **R1‑9: "Figure 3 small/unreadable; unclear seeds; Phoenix zero std; introduction ordering; afterstate values; learning vs planning; how are $T_s$ and $T_{s,a}$ tracked, transitions simulated, cached?"**
>
> * **Figures & seeds.** We enlarged the main SyntheticTree  plot, improved color contrast between CATSO and PATSO, increased label sizes, and added the number of seeds (10) and 95% CIs in the caption.
> * **Phoenix's zero std.** Phoenix behaves almost deterministically under a fixed initial state and deterministic policy + emulator seed; hence the low variance. We now explicitly note this and clarify that the main stochasticity stress‑test is the SyntheticTree environment and highly stochastic Atari games.
> * **Introduction & contributions.** We moved the contribution bullet list earlier and shortened the preamble to avoid the impression of "related work first".
> * **Afterstate values / stochastic MuZero.** We now briefly cite and discuss afterstate‑based approaches (e.g., stochastic MuZero) in the related work, explaining that our focus is complementary: we study **distributional planning** with a fixed value function rather than learning afterstate value functions.
> * **"Learning" vs "planning."** We clarified throughout that our algorithms are planners; on Atari we use a pretrained DQN purely as a feature/value provider.
> * **Implementation details of $T_s$, $T_{s,a}$, transitions, caching.** Sec. 3.2 now states more explicitly that we follow the standard MCTS pattern: for each simulation we traverse the current tree, increment visit counts ($T_s$, $T_{s,a}$) along the path, simulate transitions via the environment's generative model, and cache newly visited states as V‑nodes and Q‑edges.
>
> We appreciate these suggestions-they helped us improve both exposition and positioning. We kindly ask you to consider re-evaluating our work in light of these comprehensive revisions, and we remain open to any further suggestions you may have.

---

> ### Comment · Reviewer_Wj3L · 2025-11-27
> **Thanks**
>
> Thanks for the revised ablation, this made the paper a lot stronger.
> However,
>
> * why not full sweep in Table 3? I would recommend using a plot.
> * why $(k,d)=(8,3)$ in table 4 ? It makes sense to use $(16,4)$ or $(16,3)$ where your approach performed the best. Yours are supposed to work best, right?
>
> In table 1, how did you determine the wins and ties? statistical test?
>
> Reflecting the updates, I will raise the score, but there is still overall lack of polish in this paper like those mentioned above.
> To be honest, updating the paper like this during the review phase is not really how conferences work.
> Please submit a complete paper and don't waste reviewers' time with unpolished papers.
> That being said, I encourage the authors to keep improving the paper and I hope to see it accepted in future venues.

---

> > ### Author Response · Authors · 2025-12-01
> > **Response to reviewer Wj3L**
> >
> > Thank you again for carefully rereading the paper, for your very constructive criticism on the experiments and exposition, and for raising your score. Below we address your concrete questions and explain how we revised the paper.
> >
> > ### (1) "Why not full sweep in Table 3? I would recommend using a plot."
> >
> > You are absolutely right that plots are often the clearest way to present sweeps.
> >
> > In our SyntheticTree section, the **full sweep over tree shapes and budgets is already shown in Fig. 1**: it contains curves of root‑value error vs. $n_{\text{sims}}$ for all 40 $(k,d)$ combinations and all main algorithms, so it already plays the role of the "full sweep" over tree configurations.
> >
> > Table 3 is meant to serve a different, more compact purpose: it zooms in on **six representative $(k,d)$ pairs** at a fixed budget $n_{\text{sims}} = 1000$, and only for the four algorithms that appear in the component ablation (best distributional TS+max‑backup vs. best scalar baseline). It is essentially a *summary slice* of the information that is visualized more in Fig. 1.
> >
> > Given the 10‑page limit, adding a second multi‑panel figure that would largely duplicate the SyntheticTree curves in Fig. 1 would force us to cut other material. Our hope is that, with the explicit pointer to Fig. 1 and more experiments table results were added to the appendix, it provide more full sweeps and that Table 3 is deliberately a concise summary rather than an incomplete picture.
> >
> > ### (2) "Why $(k,d) = (8,3)$ in Table 4? It makes sense to use $(16,4)$ or $(16,3)$ where your approach performed the best."
> >
> > For the **stochasticity sweep** (Table 4) our intention was not to pick the configuration where CATSO/PATSO look the strongest, but to use a **mid‑range tree** that is representative and cheap enough to re‑run under several noise regimes.
> >
> > The choice $(k,d) = (8,3)$:
> >
> > * lies between the shallower/narrower and deeper/wider trees shown in Fig. 1,
> > * produces errors that are neither trivially small nor saturated for any method, and
> > * keeps runtime manageable when we repeat the experiments across four noise settings (deterministic, low, medium, high).
> >
> > On this configuration we see the qualitative behavior we wanted to highlight:
> >
> > * PATSO consistently matches or improves on the scalar TS+optimism baseline (ScalarTSOpt),
> > * distributional methods remain competitive with Power‑UCT even at low noise, and
> > * as noise increases, the benefits of modeling full return distributions become more pronounced.
> >
> > ### (3) "In Table 1, how did you determine the wins and ties? statistical test?"
> >
> > Thank you for raising this. We have changed the caption to make this explicit clearer.
> > * The "Wins/Ties" row is **purely descriptive**.
> > * A **win** for a method on a game is counted whenever it has the **highest mean return** among all methods (after rounding to the precision shown in the table).
> > * A **tie** is counted when the method **shares that best mean value** (again, up to rounding) with at least one other method.
> >
> > We do **not** run formal statistical tests (no $t$‑tests, Wilcoxon tests, or bootstraps) for these counts. The main information for the reader should be the per‑game mean $\pm$ standard deviation values, with the "Wins/Ties" row serving only as a compact summary of how often each method attains or shares the best mean.
> >
> > ### (4) On polish and update timing
> >
> > We take your comments on polish and "work‑in‑progress" submissions very seriously. The initial version was clearly not as polished as it should have been. For the revision we already did:
> >
> > * Rewrote Sections 1-3 to emphasize intuition and high‑level structure before technical detail, and to streamline the method description.
> > * Simplified the presentation of Q‑node updates and action selection, and moved low‑level derivations to the appendix.
> > * Removed all non‑compliant formatting (negative `\vspace`, irregular section spacing), cleaned up figures and tables, and ensured the paper adheres to the ICLR style file.
> > * Added the ablations, stochasticity sweeps, and hyperparameter sweeps you asked for, along with scripts and CSVs so the results are fully reproducible.
> >
> > We appreciate the reminder that reviewers' time should not be spent fixing unpolished drafts; this has already changed how we will prepare submissions for future venues.
> >
> > ### (5) A brief note on score
> >
> > We are very grateful that you already raised your score from 2 to 4 after our first round of revisions. Given the additional improvements we have now made-especially the expanded ablation suite, the clarified theory section, and the significantly cleaned‑up presentation-we would be honored if you felt that the current version merits a further increase in your assessment. Of course, we completely respect your judgment either way and are thankful for the time you have invested in helping us improve this work.

---

### Author Response · Authors · 2025-11-25
**To all Reviewers**

We sincerely thank all reviewers for their detailed and constructive feedback.
We have significantly revised the manuscript to address the main concerns about clarity, formatting, disentangling algorithmic components, and experimental evidence.

**(1) Formatting and structure.**
We removed all manual spacing commands (e.g., negative `\vspace`) and strictly adhere to the ICLR style. Section spacing and figure placement have been cleaned up, and the main paper now fits within 10 pages without format hacks. We restructured the introduction to present our contributions earlier and streamlined Sections 3–4 to read more like a compact conference paper rather than a technical report.

**(2) Clarity of the methods (Sec. 3).**
Sections 3.1–3.2 were rewritten to emphasize high‑level intuition before technical details:

* We now first describe, in words, how CATSO and PATSO maintain Q‑node *distributions* and perform Thompson sampling with an optimism bonus, and only then give the detailed update rules (with some steps moved to the appendix).
* We added a dedicated paragraph explaining why V‑nodes remain scalar and how the power‑mean backup fits into the overall design.
* PATSO with **merge‑on‑insert and particle cap $K$** is now consistently presented as the *default* variant in both the theory (Sec. 4.4) and experiments (Sec. 5). The uncapped version appears only as a conceptual starting point.

**(3) Theory and assumptions (Sec. 4).**
We clarified the theoretical parts as follows:

* In Sec. 4.1 we now *explicitly* state and justify the "asymptotically stationary" assumption for non‑stationary bandits in the MCTS setting: for any fixed node, as the subtree stabilizes, the rollout returns converge to i.i.d. samples from the truncated value $\widetilde{Q}(s,a)$. We connect this to the non‑stationary bandit view used in Fixed‑Depth‑MCTS and related work.
* Sec. 4.3 (Wasserstein DR‑MDP) was rewritten: all notation (including $W_p$) is defined, the robust policy is correctly written with `argmax`, and we now give a clean statement of the $(\varepsilon,\delta)$-robust policy definition and sample‑complexity result.
* Sec. 4.4 emphasizes that the capped PATSO variant preserves the $O(n^{-1/2})$ simple‑regret rate up to an additive $O(\mathrm{range}/(K(1-\gamma)^2))$ term, and we empirically confirm that this term is negligible for moderate caps.

**(4) Ablations, stochasticity, and hyperparameter sensitivity (Sec. 5.3).**
We added a dedicated *Ablations and sensitivity analysis* section on **SyntheticTree**:

* **Component ablations (Tables 2–3).**
  We now separately evaluate:

  * CATSO & PATSO with mean backup ($p=1$) vs max backup ($p\to\infty$),
  * a *scalar* TS+optimism baseline (ScalarTSOpt), and
  * Power‑UCT (UCB‑style selection + power‑mean backup, no distributional Q‑nodes).
    These ablations directly isolate the contribution of **distributional Q‑nodes** vs scalar estimates and the effect of the backup rule.

* **Effect of stochasticity (Table 4).**
  On a fixed tree $(k,d)=(8,3)$ we vary both reward and transition noise (deterministic, low, medium, high). This addresses the question: *how do distributional methods behave across deterministic and increasingly stochastic environments?* We show that:

  * in deterministic / low‑noise regimes, Power‑UCT is often best but PATSO remains competitive, and
  * as stochasticity increases, PATSO and CATSO gain a clear advantage over the scalar TS+optimism baseline (ScalarTSOpt), in line with our motivation.

* **Hyperparameter sweeps (Tables 5–6).**
  We sweep the power‑mean exponent $p$, the optimism constant $C$ (PATSO), the number of atoms $N$ (CATSO), and the particle cap $K$ (PATSO). Performance is quite flat for $p \in \{1,2,4,8\}$ and for a broad range of $C,N,K$; max‑backups ($p\to\infty$) help at higher budgets, but the sensitivity is low. These results both answer the reviewers' questions about tunability and show that our defaults (e.g., $C=1.5$, $N=100$, $K=200$) are robust.

* **Runtime.**
  We added a runtime comparison: on SyntheticTree with $(k,d)=(8,3)$, UCT is fastest, Power‑UCT incurs a small overhead, and CATSO/PATSO are ~1.5–2× slower per move in Python. We provide a script (`runtime_benchmark.py`) to reproduce wall‑clock numbers.

---

> ### Author Response · Authors · 2025-11-25
> **To all Reviewers**
>
> **(5) Experimental details and significance (Sec. 5.1–5.2 & Appendix F).**
>
> * We clarified that all SyntheticTree results use 10 seeds, report mean absolute root errors with 95% CIs, and explicitly state the simulation budget ($n_\text{sims}=1000$ unless otherwise noted).
> * For Atari, we now emphasize that we run 10 episodes per game per seed, with 3 seeds (30 episodes per method per game), and we report mean ± stdev. We added a "Wins/Ties" row to help interpret significance across games.
> * We improved the legibility of the main plots (larger fonts, clearer color schemes, more spacing between subplots) and aligned Table 1 numerically to make comparisons easier to read.
>
> **(6) Positioning, learning vs planning, and V‑nodes.**
>
> * We clarified that our contribution is **planning under uncertainty**, not a new RL training pipeline. On Atari we deliberately decouple learning and planning by using a pretrained DQN as a *fixed* value function, to isolate the effect of distributional MCTS. We rephrased this in the introduction and experimental section to avoid any confusion.
> * Sec. 3.3 and the discussion (Sec. 6) now more clearly explain why V‑nodes remain scalar: propagating full distributions through V‑nodes would require expensive cross‑child convolutions, scaling poorly with branching factor and depth and making the analysis much harder. Our choice keeps the core novelty-distributional Q‑nodes where decisions are made-while remaining practical.
>
> We hope these changes alleviate the main concerns about clarity, experimental design, and the interpretation of our results.

---

### Meta-Review · Area_Chair_gEK2 · 2026-01-06

**Summary:**

The manuscript studies Monte Carlo tree search in stochastic environment and proposes distributional MCTS algorithms with optimistic exploration bonus. While the approaches seem promising, the presentation was lacking. The authors made a significant revision of the manuscript during the discussion period. Due to the special circumstance of this year, the meta-reviewer feels that the revised manuscript should be better considered at another venue with a more thorough review process.

**Reviewer Concerns:**

The authors made a substantial revision addressing most of the reviewer comments, while the meta-reviewer feels that it needs further polishing after reading the revised version.

**Reviewer Scores:**

While two reviewers indicated that they might consider raising their ratings, I don't believe the changes will raise above the bar of acceptance.

---

### Decision · Program_Chairs · 2026-01-26

Reject